# Insulinoma-Associated Protein 1 (INSM1): Diagnostic, Prognostic, and Therapeutic Use in Small Cell Lung Cancer

**Renato Rocha** [1] and **Rui Henrique** [2,3,4,*]

1   Integrated Master in Medicine, School of Medicine and Biomedical Sciences, University of Porto (ICBAS-UP), Rua Jorge Viterbo Ferreira 228, 4050-513 Porto, Portugal
2   Department of Pathology and Molecular Immunology, School of Medicine and Biomedical Sciences, University of Porto (ICBAS-UP), 4050-513 Porto, Portugal
3   Department of Pathology, Portuguese Oncology Institute of Porto (IPO-Porto)/Porto Comprehensive Cancer Centre (P.CCC), R. Dr. António Bernardino de Almeida, 4200-072 Porto, Portugal
4   Cancer Biology & Epigenetics Group—Research Center of IPO Porto (CI-IPOP)/RISE@CI-IPOP (Health Research Network), Portuguese Oncology Institute of Porto (IPO-Porto)/Porto Comprehensive Cancer Centre (P.CCC), R. Dr. António Bernardino de Almeida, 4200-072 Porto, Portugal
*   Correspondence: rmhenrique@icbas.up.pt or henrique@ipoporto.min-saude.pt

**Abstract:** Small cell lung carcinoma (SCLC) is an aggressive and difficult to treat cancer. Although immunohistochemistry is not mandatory for a SCLC diagnosis, it might be required, especially in small samples. Insulinoma-associated protein 1 (*INSM1*) is expressed in endocrine and nervous tissues during embryogenesis, generally absent in adults and re-expressed in SCLC and other neuroendocrine neoplasms. Its high specificity propelled its use as diagnostic biomarker and an attractive therapeutic target. Herein, we aim to provide a systematic and critical review on the use of *INSM1* for diagnosis, prognostication and the treatment of SCLC. An extensive bibliographic search was conducted in PubMed® focusing on articles published since 2015. According to the literature, *INSM1* is a highly sensitive (75–100%) and specific (82–100%) neuroendocrine immunohistochemical marker for SCLC diagnosis. It can be used in histological and cytological samples. Although advantageous, its standalone use is currently not recommended. Studies correlating *INSM1* expression and prognosis have disclosed contrasting results, although the expression seemed to entail a worse survival. Targeting *INSM1* effectively suppressed SCLC growth either as a suicide gene therapy regulator or as an indirect target of molecular-targeted therapy. *INSM1* represents a valuable biomarker for a SCLC diagnosis that additionally offers vast opportunities for the development of new prognostic and therapeutic strategies.

**Keywords:** *INSM1*; biomarker; immunohistochemistry; small cell lung carcinoma; diagnosis; prognosis; therapy

## 1. Introduction

Lung cancer is the second-most prevalent and the deadliest cancer worldwide, having caused an estimated 1,796,144 deaths in 2020 [1]. Small cell lung carcinoma (SCLC) accounts for approximately 14% of lung cancer diagnoses in the US [1], characterized by highly aggressive behavior, frequent metastasis to multiple sites, and a high recurrence rate after the initial response to chemotherapy, with only 7% of patients, on average, surviving at least 5 years [1–4]. Typically, SCLC has a central perihilar location, with frequent mediastinal lymph node involvement at the time of diagnosis [5]. Its central location hampers surgical and cytological sample acquisition, which are, nonetheless, required for a definitive diagnosis [5].

The histopathologic evaluation of biopsy specimens is essential, required for appropriate tumor classification and staging. Morphologically, SCLC is composed of small cells with scant cytoplasm, finely granular ("salt and pepper") chromatin, absent or inconspicuous

nucleoli, a high mitotic count (in resection specimens, > 10 mitoses/2 mm$^2$), and abundant necrosis. In small biopsies, a high proliferation index (Ki67) is a useful finding to support the diagnosis. Two subtypes are listed: "pure" SCLC and combined SCLC, the latter having, in addition to the main SCLC component, a non-small cell carcinoma (NSCLC) component [5]. SCLC is included in the group of lung neuroendocrine neoplasms (NEN), a set of neuroendocrine tumors that also comprises the low-grade typical carcinoid (TC), the intermediate-grade atypical carcinoid (AC), and neuroendocrine carcinomas (NEC), formerly high-grade neuroendocrine carcinomas, encompassing large cell neuroendocrine carcinoma (LCNEC) and SCLC itself. Although included in this group, a SCLC diagnosis exempts the expression of neuroendocrine (NE) immunohistochemical (IHC) markers in opposition to LCNEC, which mandatorily requires the expression of one or more markers [5]. Moreover, the different subtypes have markedly diverse clinical and pathological characteristics [6]. The SCLC treatment strategy differs from other pulmonary NEN and NSCLC, entailing the use of more aggressive chemotherapy regimens [6,7]. Hence, reliable diagnostic biomarkers are the key to facilitate a SCLC diagnosis, considering the frequent challenge in obtaining histological or cytological materials [8], given the location and advanced stage at diagnosis. Furthermore, if this marker would also prove useful for prognostic and predictive therapeutic effects, it would be a substantial asset to clinical practice and to prolonging the survival of these patients.

Insulinoma-associated protein 1 (INSM1) is a 510 amino acid transcription factor with a C-terminal containing five zinc-finger DNA-binding motifs and an N-terminal exhibiting repressor activity [9,10] that performs a key role in the physiologic development of neuroendocrine tissues across the body [11–17]. It is encoded by an intronless gene localized in the short arm of chromosome 20, *INSM* transcriptional repressor 1 (*INSM1*), originally *IA-1*, first discovered in 1992 in human pancreatic insulinoma tissue and murine insulinoma cell lines by Goto et al. [9,18]. Although present in the embryonic organogenesis stage of endocrine and nervous tissues, its expression virtually disappears in normal adult tissue [10,19,20], with a few exceptions, which include Kulchitsky cells of respiratory and gastrointestinal epithelium, NE cells of the adrenal medulla, pancreatic islets, and non-neoplastic prostate gland cells [10,21–25]. As a result, *INSM1* shows an aberrant expression in adult tissues in a great variety of neuroendocrine and neuroepithelial neoplasms, including those of the lungs [4,6,13,19,21–24,26–40], pancreas [41], gastrointestinal tract [21,42], appendix [43], bladder [44,45], prostate [46], uterine cervix [47], head and neck [48], larynx [49], skin [50–52], pituitary gland [4], thyroid [53], parathyroid [21], and adrenal gland [4]. Other types of neoplasms with neuroendocrine differentiation also disclose *INSM1* expression, such as paraganglioma [35,37], extra-skeletal myxoid chondrosarcoma [54], primitive neuroectodermal tumors [21], and peripheral neuroblastic tumors [55].

Herein, we aimed to provide a systematic and critical review of the available literature on *INSM1*, focusing on its potential role as a diagnostic and prognostic biomarker, as well as therapeutic target, in SCLC, since most of the published articles tackle its role in neuroendocrine tumors in general (both in the lungs and other tissues). We believe that summarizing the current knowledge on *INSM1* in SCLC in a single article will enable a clearer view and understanding of its potential by pathologists and clinicians.

## 2. Methodology

An extensive bibliographic search primarily in the PubMed$^®$ database, using the following key terms: "INSM1", "small cell lung cancer", "SCLC", "Immunohistochemistry", and "biomarkers", took place alone or in different combinations. The terms "CD56 Antigen", "Synaptophysin", "Chromogranin A", and "Molecular targeted therapy" were also secondarily used.

The main selection criterion was articles published in the past 6 years, since it was in 2015, the publication date of Rosenbaum et al. and Fujino et al.'s articles [4,21], that INSM1 began to be widely used as an immunohistochemical marker. Articles outside of this time range were included when deemed relevant. We selected exclusively articles in

the English language, with full text available and intrinsically related to the review's theme, containing the terms previously mentioned by first checking the title and abstract and then, secondarily, the body of the article.

For the table construction, we included any article that specified INSM1 immuno-chemical use in previously diagnosed SCLC and had more than 10 SCLC samples analyzed. Studies that used *INSM1* in high-grade neuroendocrine tumors but did not discriminate SCLC from other types of cancer were excluded. Microsoft® Excel® for Microsoft Office 365® (Microsoft Corporation, Redmond, WA, USA) was the software used to construct these tables.

Original figures were constructed using Microsoft® PowerPoint® for Microsoft Office 365® (Microsoft Corporation, Redmond, WA, USA).

### 3. INSM1 and SCLC Oncogenesis

*INSM1* is a key player in various biological functions and interacts with a wide range of molecules. It has an essential role in embryogenesis and differentiation of the pancreas, sympathoadrenal system, and nervous system [10,11,56,57]. It partakes in distinct signaling pathways, including the Sonic Hedgehog (*Shh*), Phosphatidylinositol-4,5-Bisphosphate 3-Kinase (*PI3K*)/AKT serine/threonine kinase (*AKT*), Mitogen-Activated Protein Kinase Kinase (*MAP2K*)/ Extracellular Signal-Regulated Protein Kinase (*ERK*)$^{1/2}$, adenosine kinase (*ADK*), Tumor Protein 53 (*TP53*), Wingless-Type *(Wnt)*, histone acetylation, Lysine-Specific Demethylase 1 (*LSD1*), cyclin D1, achaete-scute family BHLH transcription factor 1 (*ASCL1*), and *MYCN* proto-oncogene bHLH transcription factor (*MYCN*) pathways [58,59]. These signaling pathways are not uniformly present throughout the body, and *INSM1* interactions with various molecules in neuroendocrine cells of different locations are due to multiple tissue-specific regulatory elements in the upstream region of the gene [18]. Since it is not the focus of this review and has been already covered elsewhere [10,58,59], the overall function of *INSM1* is described here in a simplistic manner.

*INSM1* exhibits activity as a transcriptional repressor, binding to genes such as NeuroD/β2 and insulin gene (*INS*), fundamental to pancreatic development, while simul-taneously exhibiting extranuclear activity through binding to cellular regulator proteins such as receptor for activated C kinase 1 (RACK1) and cyclin D1, promoting differentiation and cell cycle arrest [60]. Via interactions with Neurogenin-3 and neuronal differentiation 1 (*NEUROD1*), *INSM1* participates in pancreatic β-cell differentiation, which illustrates one of its functions as a crucial regulator of neuroendocrine differentiation [10]. INSM1 binding to Phox2b and *ASCL1* promotes sympathoadrenal cell differentiation, and, in a study with mice, a homozygous mutation of the gene was shown to cause fetal death due to catecholamine deficiency [57]. In the brain, *INSM1* plays a pan-neurogenic role and fosters basal progenitor cell formations, supporting cortex development and expansion [56,61]. One of its most compelling functions is its autoregulation capacity, through binding of the *INSM1* protein to the promoter region of its own gene [16], with studies showing that this region is a possible target for cancer gene therapy [62–68]. Furthermore, it contributes to the differentiation of anterior pituitary and enteric endocrine cells, the transformation of neuroblastoma cells, and, possibly, when decreased, to the development of diabetes mellitus [11,58,69–71].

INSM1 activity is also apparent in the lungs, as pulmonary neuroendocrine cell differ-entiation is dependent on complex interactions between *INSM1*, *ASCL1*, and Hes Family BHLH Transcription Factor 1 (*Hes1*) [4,12]. During lung organogenesis, epithelial cells may take two routes: neuroendocrine differentiation or non-neuroendocrine differentiation [72]. *Hes1*, a Notch-signaling gene responsible for inhibiting the formation of neuroendocrine cells, is expressed in non-neuroendocrine cells, while *ASCL1,* a master regulator of neuroen-docrine differentiation, is expressed in neuroendocrine cells [12]. Thus, *ASCL1* expression is negatively correlated with Hes1 activity [12]. A murine model showed that *INSM1* mu-tants depicted *ASCL1* downregulation and *Hes1* upregulation, compromising the terminal neuroendocrine differentiation, and that INSM1 could bind to the *Hes1* gene in different

sequences, including a promoter region, repressing its transcriptional activity and consequently allowing for neuroendocrine cells differentiation [12]. Epigenetic mechanisms such as the recruitment of histone-modifying factors via a Snail/Gfi-1 (SNAG) domain contained in the *INSM1* N-terminus may also play a part in repressing *Hes1* expression [12,69]. On the other hand, *ASCL1* seems to directly or indirectly control *INSM1* expression, suggesting a possible interdependence [12]. Subsequently, a study with a conditional lung-specific *INSM1* transgenic mouse model illuminated the INSM1 effects on lung alveologenesis, revealing that, when ectopically expressed on bronchiolar epithelial cells, alveolarization is compromised in the terminal stages of lung maturation, leading to septation defects and, ultimately, alveolar space enlargement [25]. This process resulted from the inhibition of cyclin D1 expression in club cells, with the consecutive blockage of bronchiolar epithelial cell cycle progression [25]. Club cells do not normally express *INSM1* but have been found to be closely connected with pulmonary neuroendocrine cells, with the latter acting as stem cells and a means of regeneration for club cells in the event of injury. Thus, *INSM1* might be involved in lung injury repair pathways [25,73]. Interestingly, in this study, *INSM1* ectopic expression in non-NE cells did not induce NE precursor differentiation, implying the existence of alternative molecular pathways [25]. In short, pulmonary neuroendocrine cell differentiation is partly dependent on *INSM1* expression, whereas, in other types of respiratory cells, it may harm lung alveologenesis.

Regardless of decades of study, SCLC oncogenesis remains poorly understood. Other than the inactivation of the tumor suppressor genes *TP53* and *RB1*, no major biomolecular events are widely recognized in the genesis of this neoplasm, in addition to being vastly heterogeneous, owing to the multitude of genes potentially involved [40,74,75]. SCLC and NSCLC are clearly distinct on every level, from the clinical to histopathologic features and, accordingly, on a molecular basis [40,74]. Identifying genes with a discordant expression between these two subtypes of cancer is of great importance, as it may be useful for diagnostic, prognostic, and therapeutic purposes and to increase the knowledge about causative molecular mechanisms. Naturally, as a neuroendocrine neoplasm, the profiling of genes with a neuroendocrine function has been carried out, with many showing upregulation, including chromogranin A (*CGA*), *ASCL1*, and *INSM1*. [40,74]. *ASCL1* is reportedly associated with SCLC carcinogenesis, with studies showing a close relation between *ASCL1* and *INSM1*, since both show similar expression patterns in fetal lung neuroendocrine cells and cancer cells, exhibiting neuroendocrine differentiation [4,40,76]. Therefore, *INSM1* may be directly or indirectly implicated in SCLC oncogenesis. *INSM1* has been shown to regulate neural cell-specific transcription factor POU Class 3 Homeobox 2 (*POU3F2*), alongside *ASCL1*, in SCLC cell lines, which sequentially promote the expression of NE molecules such as CGA, synaptophysin (SYN), and neural cell adhesion molecule 1 (NCAM1) [4,38]. Previously, it was shown that *INSM1* could bind to *Hes1* in normal pulmonary neuroendocrine cells, promoting neuroendocrine differentiation [12]. Inversely, Fujino et al. showed that the forced expression of *Hes1* or Notch receptor 1 (*NOTCH1*) resulted in the inhibition of *INSM1* expression and, consequently, of the downstream molecules [4]. On another note, in that same study, *ASCL1* expression did not control *INSM1* expression, as opposed to the findings of a contemporary study by Jia et al. in normal pulmonary neuroendocrine cells, as discussed beforehand, but also contrarily to a more recent study using SCLC cell lines, where *INSM1* was upregulated by *ASCL1* [4,12,77]. Thus, the exact location of *INSM1* in *ASCL1*-related molecular pathways is not yet fully understood, and further studies are required. Fujino et al. also reported that *INSM1* expression had opposite effects in SCLC and NSCLC, with *INSM1* knockdown inducing apoptosis and decreased cell growth in the former and forced expression causing inhibited cell growth in the latter, suggesting that *INSM1*-related oncogenicity mechanisms may be dependent on other molecules exclusive to a cancer subtype, as may be the case for other neuroendocrine factors [4]. More recently, Chen et al. found that *INSM1* was involved in the complex Shh-*PI3K/AKT-MYCN/ASCL1-MEK/ERK*$^{1/2}$ transcriptional network in NE lung cancers, including SCLC [77]. *Shh* and *MYCN* are both important for normal embryogenic development but were also associated

with neuroendocrine lung tumors and other types of cancer, including neuroblastoma, in which *INSM1* may also play an active role [70,77]. Essentially, *Shh* activates *INSM1* through the activation of *MYCN* via *PI3K/AKT* and alternative pathways, which, in turn, phosphorylates *MEK/ERK^{1/2}*, ultimately resulting in the stabilization of *MYCN*, which causes further *INSM1* activation, leading to a positive feedback loop that promotes tumor growth [77]. The upregulation of *INSM1* is caused by the binding of MYCN and ASCL1 to an E2-box sequence in its promoter region [77]. In summary, *INSM1* is crucial for the NE differentiation of SCLC, associated with various master NE genes, such as *ASCL1* and *Shh*, in pathways most likely connected to oncogenesis. The *INSM1* intervention in SCLC oncogenic molecular pathways is displayed in Figure 1.

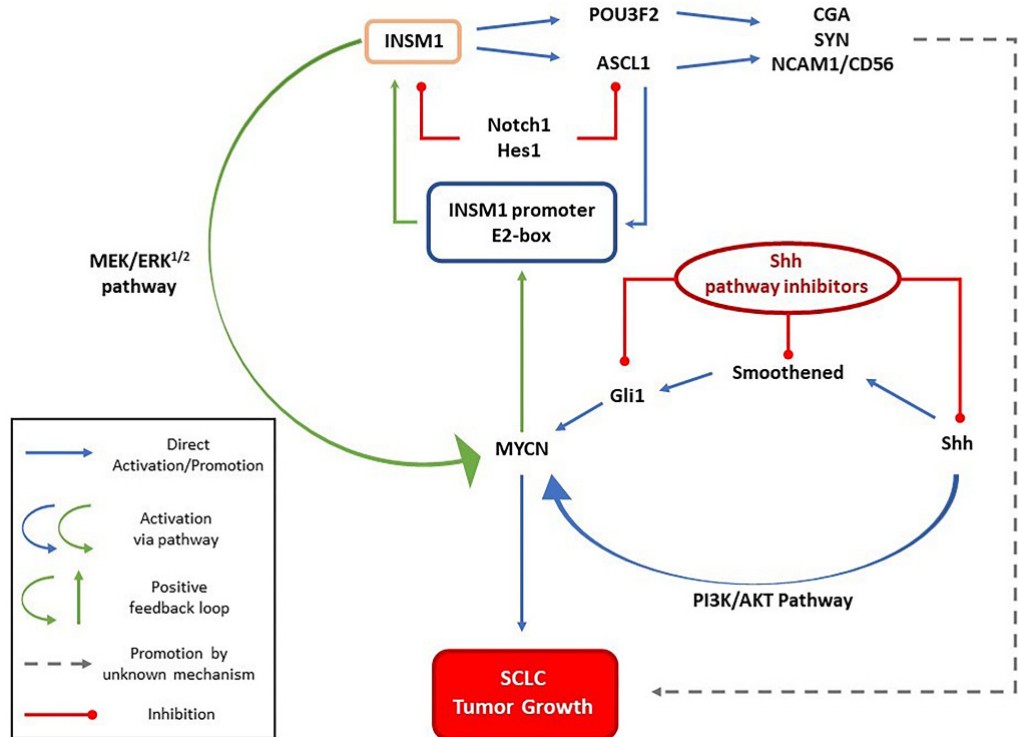

**Figure 1.** Intervention of *INSM1* in SCLC oncogenic molecular pathways. *INSM1* participates in different molecular pathways that may promote SCLC oncogenesis. *INSM1* regulates *POU3F2* alongside *ASCL1* in SCLC cell lines, which sequentially promote the expression of NE molecules such as CGA, SYN, and NCAM1/CD56, which have been shown to promote SCLC tumor growth through unknown mechanisms [4,38]. *Notch1* and *Hes1* inhibit both *INSM1* and *ASCL1* [4]. *MYCN* and *ASCL1* can bind to the *INSM1* promoter E2-box, upregulating *INSM1*. Additionally, *MYCN* has been shown to promote SCLC tumor growth [77]. *INSM1* can activate *MYCN* via the *MEK/ERK^{1/2}* pathway, which then binds to the *INSM1* promoter E2-box, causing further *MYCN* activation and forming a positive feedback loop (in green) that promotes tumor growth. *MYCN* is also activated by Shh either by the *PI3K/AKT* pathway or *Shh* pathway [77]. The inhibition of SCLC tumor growth was achieved by the blockage of the Shh pathway using different inhibitors [77].

## 4. INSM1 Diagnostic Use

### 4.1. Performance of INSM1 as a Diagnostic Biomarker

There are multiple techniques for detecting *INSM1* expression in different types of samples. The vast majority of published reported focusing on immunohistochemistry (IHC), and thus, this will be the main emphasis of this review. Nonetheless, because there are some studies on immunocytochemistry (ICC) [22,29–34,36] and microarrays or PCR (polymerase chain reaction) of *INSM1* mRNA [21,28,39,62,74,78], these techniques will also be covered here.

### 4.2. Immunohistochemistry

The diagnosis of SCLC does not absolutely require immunohistochemistry, as it may be achieved by histomorphological observation alone [5], but it improves the diagnostic accuracy [8], particularly in cases with limited materials, such as small biopsies, a frequent type of sample in this scenario, and consequently, a reliable immunohistochemical marker is very important. Classic neuroendocrine markers such as CGA, SYN, and CD56 are widely used by pathologists, but all of them have limitations, with CGA typically disclosing the lowest median sensitivity, ranging from 46–100% [4,6,13,23,24,26,28,32,35–38], SYN depicting an intermediate sensitivity ranging from 55–100% [4,6,13,23,24,26,28,32,35–38], and CD56 typically showing the highest sensitivity, ranging from 68–100% [13,23,24,26,28,31,32,35–38]. On the other hand, CGA discloses the highest specificity, and all of them are prone to showing nonspecific cytoplasmatic and membranous background staining, particularly in areas with necrosis, which are common in small biopsies, making a diagnosis difficult, especially in cases with weak or focal expression. Additionally, 10–25% of lung NEC, similar to SCLC, might not stain with the typical triple-marker panel [5,30,79,80]. Indeed, searching for a marker that might be highly sensitive and specific is a priority, as replacing multiple marker panels with a single one should not only save tissue, which is frequently scarce in the context of SCLC and lung pathology in general with the advent of molecular studies [27], but also reduce costs [23,37]. In recent years, interest in *INSM1* as a diagnostic NE biomarker for IHC has grown after the initial studies by Rosenbaum et al. and Fujino et al. [4,21]. We will now proceed to discuss in detail the main studies on INSM1 IHC use for SCLC diagnosis.

The study of Rosenbaum et al. was the first to report INSM1's utility as a diagnostic biomarker in a large set of neuroepithelial neoplasms. It demonstrated that *INSM1* is exclusively expressed in NE tumors and some types of NE cells in adults, such as adrenal medulla cells, pancreatic islet cells, gastrointestinal enterochromaffin cells, and, rarely, small nests in the bronchial epithelium and individual prostatic glandular cells, whereas the expression was absent in a large series of normal non-neuroendocrine non-neoplastic adult tissues [21]. Although INSM1 is expressed in some normal NE cells, these may be easily distinguished from neoplastic tissue, representing only a minor drawback. Importantly, its expression was observed in 88.3% of NE neoplasms, including all NE tumor types surveyed, except for parathyroid adenomas and carcinomas. This finding deserves more studies to explore whether alternative molecular pathways may justify the absence of expression [21], eventually having importance for diagnosis and therapy. Although the sample set included only three cases of SCLC, all stained positive, attesting the importance of this study as a starting point for further research on this topic [21]. The fact that the expression was nuclear with minimal background staining also made this marker an attractive target for future studies [21].

Remarkably, in the same year, Fujino et al. published a study focused on the INSM1 neuroendocrine role in lung cancer, reporting 100% (27/27) sensitivity for INSM1 IHC with also a perfect specificity in differential diagnosis with NSCLC, as its negative expression was disclosed in 86 and 47 cases of adenocarcinoma (ADC) and squamous cell carcinoma (SCC), respectively [4]. Interestingly, eight cases corresponded to combined SCLC-NSCLC, with INSM1 expression only detected in the SCLC component, emphasizing that *INSM1* is mostly, if not exclusively, expressed in tumors and cells bearing NE differentiation [4,21]. Both CGA and SYN performed worse than INSM1—70% (19/27) and 63% (17/27) sensitivity, respectively. Moreover, a Western blot analysis was performed, and although the protein expression was not universal in the SCLC cases tested (4/7), it was not observed in NSCLC [4]. These findings highlight the potential of INSM1 to differentiate SCLC and NSCLC in difficult cases, considering perfect specificity, as well as the ability to confirm mixed SCLC-NSCLC cases, which may be candidates for alternative treatment strategies in the future [5].

In a subsequent publication, the same authors again reported 100% sensitivity, this time assessing 19 cases of SCLC but with the added value of a comparison with CD56, which depicted 68% (13/19) sensitivity, less than reported in other similar studies [13,23,35,38],

possibly due to the smaller sample size. Contrarily to other studies but in line with their previous one, SYN and CGA disclosed a lower sensitivity (58% (11/19) and 74% (14/19), respectively), probably due to the tissue preparation protocol used [4,38]. The specificity for the differential diagnosis with NSCLC remained 100%, and the authors denoted a positive expression in all LCNEC (four cases) and carcinoid tumors (five cases) assessed. Interestingly, the authors reported that cases with a low expression of the classic NE marker disclosed diffuse INSM1 expression, and two cases of SCLC were uniquely positive for *INSM1*. Furthermore, INSM1 was shown to be superior to other NE markers not only in terms of sensitivity but also by showing a significantly higher *H*-score—246 vs. 87, 84, and 38 in CGA, SYN, and CD56, respectively. The reproducibility of *INSM1* IHC was confirmed as a comparable expression was disclosed for automated IHC protocols performed in two other laboratories [38]. Integrating a high sensitivity, specificity, and reproducibility, the authors concluded that INSM1 was clearly superior to other NE markers, larger-scale studies with a similar design would be required to corroborate their results.

In a later study, Rooper et al. reported 95% (37/39) sensitivity for INSM1 IHC in SCLC, better than all three classic NE markers—49% (19/39) CGA, 62% (24/39) SYN, and 70% (21/30) CD56—and even better than their combined sensitivity, 74% (29/39), with a mean intensity of 2.4/3, staining, on average, more than 50% of all tumor cells, with only two cases showing focal staining [37]. Remarkably, eight SCLC were INSM1 positive, but negative for all the other markers but with less intensity and percentage of positive tumor cells than tumors that expressed other NE markers. The specificity for the differential diagnosis with NSCLC reached 100%, and the expression was high in other lung NE tumors—91% (21/23) for LCNEC and 100% (48/48) for carcinoid tumors. Paragangliomas also disclosed 100% (8/8) *INSM1* expression. These promising findings led the authors to recommend the replacement of multi-marker panels with INSM1 [37].

In a large study, comprising 144 NE tumor samples, conducted by Kriegsmann et al., the *INSM1* performance was somewhat disappointing, with 86% (124/144) sensitivity for SCLC, only significantly superior to CGA (74%) and markedly worse compared to the combined panel of classic NE markers (95%) [35]. For the differential diagnosis with NSCLC, 99% specificity was achieved, and intriguingly, the expression in other NE tumors was lower than in previous studies—42% for LCNEC and 79% for carcinoids—which might be due to the larger sample size. The intensity of the positive tumor cells in SCLC was mostly medium-strong, as well as diffuse (median expression of 82% of the total cells). This study also included paragangliomas, which disclosed INSM1 IHC results similar to SCLC concerning the extension and intensity of the expression but also exhibited a higher sensitivity—97% (29/30). These results emphasize that INSM1 may be expressed by other NE tumor types, and interpretations should be careful. Kriegsmann et al. concluded that INSM1, regardless of its potential role as an additional marker for SCLC and NE differentiation in general, should not replace the conventional NE markers, as recommended by others [24,37].

The study of Švajdler et al., on the other hand, aimed to compare *INSM1* immunochemistry to a combined panel of CD56, p16, and transcription termination factor 1 (TTF1) for SCLC diagnosis, achieving 81% (81/100) sensitivity in histological samples and 54% (7/13) in cell blocks [31]. The panel demonstrated a statistically superior sensitivity, correctly identifying all 100 cases of SCLC in the histology samples, with CD56 alone disclosing the best sensitivity (84%), although non-statistically significant, and identified all 13 cases in the cytology samples, a considerably better performance, despite the small number of cases tested [31]. Staining with other classic NE markers was not performed in this study, but in another report from the same authors, the sensitivity was inferior to that of INSM1 [81]. In terms of histology–cytology correlation, INSM1 depicted the lowest concordance of positive cells among all markers, also disclosing a modest agreement between positive/negative classification (three discordant cases among nine paired specimens) inferior to that of other markers. Thus, once again, INSM1 seems to perform better in histological than in cytological specimens [31]. Indeed, cytology's small sample size may limit the interpreta-

tion of the results, a common limitation seen in other studies [30–32]. Among metastatic SCLC, 87.5% (7/8) of the cases stained positively for *INSM1*, suggesting that this marker is conserved in the metastatic setting [31]. Similar to other studies, Švajdler et al. recommend a further evaluation of the INSM1 + CD56 combination [28,30,31], but considering that the performance of this combination was still inferior to the initial panel, the authors concluded that INSM1 should be used first, and if negative, CD56 should be performed, and then, if both were found negative, TTF1 and p16 should be performed. This combination disclosed a very high negative predictive value, but it did not replace the need for a morphologic evaluation, as none of those markers are specific for SCLC vs. other NE tumors [31,37].

A contemporary study from Mukhopadhyay et al. disclosed 98% sensitivity for *INSM1* IHC (63/64), superior to CGA (83%), although similar to CD56 (95%) and SYN (100%) [23]. The staining was moderate to strong in all cases, but in 16% of them, the staining was very focal, only present in ≤10% of tumor cells. Interestingly, 80% of SCLC cases were positive for all markers, and no cases of isolated INSM1 expression were observed. There were two cases of combined SCLC-NSCLC, both disclosing INSM1 expression only in the SCLC component. For a differential diagnosis with NSCLC, 97% specificity was achieved, with positive NSCLC cases demonstrating a very focal expression, and the sensitivity in other lung NE tumors was similar for carcinoids (98%–63/64) but inferior for LCNEC (75%–18/24). The specificity was similar for CGA (98%) and superior to SYN (90%) and CD56 (87%). In three cases, primary SCLC were compared to matched metastasis, and in all cases, *INSM1* was positive, with a similar or higher *H*-score, which is in accordance with other studies [23,31]. The authors also denoted INSM1 staining in Kultchitsky cells, a type of normal NE cells of the respiratory tract, but not in other types of respiratory cells. Nonetheless, several criticisms were made of the isolated use of this marker, including (1) the presence of SCLC cases with focal expression, (2) the imperfect sensitivity, (3) the lack of SCLC positive for only INSM1, (4) the expression of INSM1 in NE tumors of many sites [21], and (5) that INSM1 does not discriminate SCLC from other lung NE tumors [23]. Thus, and contrarily to other authors [24,37], they advised the use of INSM1 only as a complementary marker [23].

On the other hand, Tsai et al. reported 83% sensitivity for INSM1 IHC among 48 SCLC cases from biopsies and resections, with 84% specificity vs. NSCLC and moderate-to-high expression in other lung NE tumors. Most INSM1+ SCLC cases displayed diffuse and strong staining patterns. This was accomplished using an ideal *H*-score diagnostic cutoff of 50 to distinguish SCLC from morphologic mimics (poorly differentiated adenocarcinomas, basaloid squamous cell carcinomas, Ewing sarcomas, small round cell tumors, and others), determined through a receiver operating characteristic (ROC) curve analysis [19]. The inferior sensitivity compared to other studies may be explained by the stricter cutoff, as most studies use any nuclear staining as the criterium for positivity [4,6,23,31,32,36–38]. Furthermore, half of the SCLC cases selected were negative for chromogranin and synaptophysin, half of them negative for TTF-1, which is a marker that, when present, makes SCLC less likely to stain for traditional NE markers and, consequently, less likely to stain with INSM1 due to the intrinsically reduced neuroendocrine phenotype [19,82]. Interestingly, 79% (19/24) of CGA/SYN-negative SCLC were positive for *INSM1*. Moreover, INSM1 expression among morphologic mimics was relevant, including 17% (4/23) of Ewing sarcomas. Where this may represent, to some extent, a bona fide NE differentiation, the main importance of these findings is to alert for possible false positives when interpreting poorly differentiated tumors similar to SCLC, once again reinforcing the need for a careful morphologic evaluation [19]. CD56 expression was also evaluated in *INSM1*-negative SCLC, and 75% (3/4) were positive, supporting the need for multiple NE markers. Some tumors expressed *INSM1* below the defined *H*-score, which means that the focal and weak expression is not enough for a SCLC diagnosis. Thus, *INSM1* should be interpreted by integrating other findings (morphology, clinical parameters, and imageology), but its value in differential diagnosis is clear, particularly when scarce material is available, as in small biopsies and cytology samples [19,30].

In the study by Staaf et al., 92% sensitivity was determined for INSM1 IHC at any nuclear staining in the tumor cells but reduced to 75% when the "at least 10% positive tumor cells" cutoff was applied. The specificity for discrimination from NSCLC reached 98%, but a high expression in other lung NE tumors was also noticed. INSM1 sensitivity was superior to CGA but inferior to SYN (79%), with the "at least 10% positive tumor cells cutoff", and CD56 (96 and 88%) at any cutoff, despite comparisons being limited by the small number of SCLC cases available. The authors found through ROC analysis that the best cutoff of *INSM1* and the other markers (except for SYN) for discrimination from NSCLC was "any number of positive tumor cells", although these results included LCNEC. When the "at least 10% positive tumor cells" cutoff was applied, none of the cases depicted isolated INSM1 positivity, and there was one combined CD56/*INSM1*-positive case [28]. The use of tissue microarray (TMA) in this study may have impacted the results, owing to tumor heterogeneity, but TMA are widely used in this type of study, because they allow for the evaluation of a larger number of cases, with reduced costs, and which would be more difficult to perform in small biopsies and cytology samples [13,24,26,28]. *INSM1* in combination with CD56 achieved the best performance among the possible combinations of NE markers [28]. The results of this study highlight the importance of defining an immunostaining cutoff value for considering a given case as positive, a definition that has not reached a consensus in the literature [24] but that may vary, depending on institution default staining protocols and pathologist's experience [30].

In the differential diagnosis of SCLC vs. NSCLC using INSM1 IHC, Sakakibara et al. observed 92% sensitivity, 95% specificity, an 84% positive predictive value, and a 97% negative predictive value in a relatively large set of SCLC (78 cases) [24]. The high negative predictive value is very important, as *INSM1* negativity can be used to rule out SCLC with a high likelihood. The INSM1 expression in LCNEC was moderate (68%) and very high (95%) in carcinoid tumors, although only 19 carcinoid cases were available. For SCLC, INSM1 sensitivity was superior to all other NE markers and even to the combined panel (85%), testing positive in 9/12 (75%) cases of SCLC negative to all classic NE markers and in 64/66 (97%) cases positive for at least one NE marker. In whole slides of those cases, a partial expression was observed in three cases considered negative in the TMA analysis, reinforcing the usefulness of *INSM1* in SCLC with none or scarce expression of the conventional NE marker [24]. This study demonstrates that *INSM1* may be used as a standalone marker for SCLC, given its high sensitivity, specificity, negative predictive value, and expression in cases testing negative for other NE markers [24].

The results from the study of Narayanan et al. disclosed 100% sensitivity (19/19) for *INSM1* IHC, superior to CGA (80%) and equal to SYN. In carcinoid tumors, 95% of the cases tested positive, an important finding considering the relatively large sample set, comprising 87 typical and atypical carcinoids. The sensitivity in LCNEC, despite being high—100%—was limited by the small number of cases (nine) [6].

In the study by Zombori et al., 100% sensitivity for INSM1 IHC was depicted in 29 cases of SCLC assessed in TMA, but the specificity vs. NSCLC was only 82%, lower than other studies, with observed INSM1 expression in 2/20 adenocarcinomas and 5/18 SCC. It is noteworthy that these cases typically disclosed a focal expression (≤75% positive tumor cells) and mild-to-moderate staining intensity. Thus, adjusting the expression criteria, they might have been considered as "negative", which emphasizes the need for strict cutoff values that may accurately discriminate significant from nonsignificant expression. *INSM1* expression was also high in other pulmonary neuroendocrine tumors [13]. The positivity was mainly diffuse (median: 95% positive tumor cells), and the staining intensity median was moderate, with an exclusively nuclear pattern, easier to interpret than other markers, in parallel with other studies [21,22,29,33–35,37]. All other classic NE markers achieved inferior sensitivity and fewer positive tumor cells, except CD56. Two cases of SCLC (histomorphologic diagnosis only) negative for all classic NE markers were found, both of which were positive for INSM1 [13]. A NE marker present when others are negative is especially valuable for SCLC considering the high frequency of low-quality samples due to

the reduced amount of tissue or crush artifacts [23,26,30]. Zombori et al. also assessed the expression of Syntaxin 1, another neuroendocrine marker with a high sensitivity and specificity for neuroendocrine neoplasms, including SCLC, and recommended the combined use of INSM1 and Syntaxin 1 (dual staining), as they have different staining locations (syntaxin stains the cell membrane), with the advantage of sparing biopsy material and allowing for easier interpretation than the classic panel of CGA, SYN, and CD56. Nonetheless, this study was limited by the low case number, use of TMA (possibility of tumor heterogeneity) and the use of classic neuroendocrine markers for the diagnosis of the selected cases, possibly interfering with their sensitivity [13]. Future studies are needed to evaluate the added value of Syntaxin 1 in combination with INSM1.

Differently from most of the studies discussed above, the study by Wang et al. did not deal with *INSM1* expression in SCLC but explored the issue of uncommon *INSM1* expression in NSCLC [27] instead, as previously reported [23,28,35,37]. Among 324 NSCLC surveyed, comprising ADC, SCC, and 42 other types (carcinoids and LCNEC excluded), the INSM1 immunohistochemical expression was positive (expression in more than 10% positive tumor cells) in 20 (6.2%), focal (expression in less than 10% positive tumor cells) in 16 (4.9%), and negative in 289 (88.9%) cases. The INSM1 expression was more common in SCC compared to ADC (9.3 vs. 5.1%), contrarily to previous publications and probably due to a higher prevalence of SCC with basaloid features [27]. Furthermore, *INSM1* was more frequently positive in poorly differentiated cases [27]. The authors repeated *INSM1* staining in surgical specimens from eight randomly selected negative cases, maintaining the same result. Moreover, repeat IHC was performed in surgical specimens available from positive and focal cases, with eight focal cases remaining focal and 5/11 positive cases in TMA changing to focal in surgical specimens, a result that emphasizes the relevance of taking tumor heterogeneity into account. SYN IHC was performed in positive and focal cases, with no expression seen in the positive cases but observed in one INSM1 focal case. This suggests that INSM1 staining in those cases may be nonspecific instead of representing true NE differentiation and supports the combined use of both markers to improve the specificity, as previously recommended [27,28]. Wang et al. emphasized that *INSM1* expression in NSCLC is mostly a problem for a differential diagnosis with SCC, as ADC very rarely shows neuroendocrine morphologic features and thus seldom requires the use of NE IHC markers [5,27]. Poorly differentiated SCC with basaloid features, however, may mimic lung NEC, including SCLC, and if INSM1 is used alone, it may lead to a misdiagnosis [27]. A further limitation is the lack of consensus regarding a cutoff value for case positivity, in agreement with others [27,28,33], albeit they used a cutoff of more than 10% immunostained tumor cells, which means that even more NSCLC could be considered positive for *INSM1* using lower cutoffs [27]. This study demonstrates that *INSM1* results need to be carefully considered in the differential diagnosis between neuroendocrine tumors (including SCLC) and NSCLC, especially in limited samples. In view of the tumor heterogeneity, it would be advisable not to use it as a standalone marker but only in combination with other NE markers and/or specific lineage markers such as p40 [27].

Yu et al. reported inferior sensitivity for INSM1 IHC—75% (59/79)—compared to SYN and CD56 (82 and 88%, respectively), although the differences were statistically nonsignificant. For the differential diagnosis of SCLC vs. NSCLC, 84% specificity was disclosed, and a moderate expression in other NE lung tumors was also observed. Importantly, among SCLC negative for classic NE markers, 53% (9/17) were positive for *INSM1*+ [26]. Although an immunohistochemically confirmed NE phenotype might not be mandatory for a SCLC diagnosis, as it may rely on morphology alone [5], it is widely acknowledged by pathologists that immunohistochemical confirmation of the NE phenotype allows for a more confident diagnosis. Thus, *INSM1* may be of great importance in that setting, as well as in cases for which classic NE markers are negative and crush artifacts, a few cells, extensive necrosis, weak/equivocal staining, or other characteristics hinder an accurate morphologic interpretation [23,26,30,36]. Yu et al. ascribed the lower sensitivity of INSM1 to different experimental conditions [26], although the differences might also be

due to the higher percentage of positive tumor cell requirements for considering a case as INSM1-positive [19].

In conclusion, INSM1 is a reliable immunohistochemical marker of NE differentiation for use in SCLC diagnosis, with high sensitivity and specificity, as demonstrated by most studies [4,6,23,24,32,37,38], with the great advantage over other markers of showing exclusive nuclear expression [13,21,35,37]. Furthermore, staining apparently represents true NE differentiation, as studies reported its expression only on the SCLC component of mixed SCLC/non-neuroendocrine tumors [4,23]. Nonetheless, INSM1 also shows expression in other neuroendocrine tumors, including carcinoid tumors of the lungs and LCNEC [6,13,19,23,24,26,28,35,37,38], and thus cannot be used to differentiate SCLC from those tumors, a task that still mostly relies on histomorphological interpretation [19]. Importantly, INSM1 expression is retained in metastases [23,31], which is of special importance for SCLC, as metastatic deposits are commonly more accessible for biopsy than the primary tumor. Some studies have found that INSM1 may be expressed in NE tumors in the absence of other NE markers [13,24,26,37,38], whereas others disclosed a superior sensitivity of INSM1 IHC, even compared to the combined panel of classic NE markers, supporting the standalone use of *INSM1* [24,37]. Nonetheless, other studies showed inferior sensitivity compared to the combined panel [23,28,31,35], no cases of isolated *INSM1* expression [23,28], or depicted the diagnostic pitfall of INSM1 expression in rare cases of NSCLC, favoring a more conservative use of *INSM1* only as a complementary NE marker [19,23,27,31,35]. Two common problems are reported in the literature: (1) the use of TMA as the type of tested sample [13,27,28], as SCLC samples are difficult to obtain and there was a need to preserve tissue, causing a possible underrepresentation of tumor heterogeneity, and (2) the lack of a standard cutoff for INSM1 IHC positivity [24,27,28], making it difficult to ascertain and compare biomarker sensitivity and specificity across studies. There is no doubt, however, of INSM1's importance as an immunohistochemical marker for SCLC, although more studies might be required to illuminate the uncertainties that remain.

*4.3. Immunocytochemistry*

As previously stated, SCLC is a tumor mainly localized in the central area of the lungs, making access to a biopsy particularly difficult. Hence, cytological samples (through FNA—fine needle aspiration) and small biopsies are, frequently, the only sources of tumors available. The former discloses a better morphological preservation of cells compared with the latter, which frequently has crush artifacts, likely making cytology samples preferable for the primary morphological diagnosis [5,29,32,34,83]. FNA also has the advantage of being minimally invasive and more cost-effective [29]. Often, FNA is the first step in the diagnostic workup of a lung mass [30,84]. With limited materials available, the immunocytochemical confirmation of SCLC through a combined panel of classic neuroendocrine markers, such as those used in IHC, can be difficult, and thus, finding a single marker constitutes a priority [4,30,32,37,38]. Recently, several studies on INSM1 immunocytochemical application were published, mostly evaluating cell blocks (CB) and direct smears [22,29,31–34,36]. Cell blocks, although preferable to direct smears, owing to their similarity to paraffin-embedded sections of biopsies, are not available in many cases, because FNA may only harvest a small number of cells [29]. Furthermore, markers with nuclear staining, such as INSM1, are preferable for direct smears [29,85,86]

The study of Doxtader and Mukhopadhyay was the first published on the usefulness of INSM1 in cytology samples, reporting a 93% sensitivity (38/41) significantly superior to CGA—35% (14/40), equal to SYN—93% (37/40), and slightly inferior to CD56—100% (40/40), disclosing 100% specificity for the differential diagnosis with NSCLC and having high expression in other lung NE tumors, although only a small number of these cases were assessed [36]. Staining for INSM1 was mostly diffuse, but 24% (9/38) of positive cases depicted a focal positivity. There were no cases with exclusive INSM1 expression, and the three *INSM1*-negative cases were CD56- and SYN-positive. The *INSM1* ICC and

IHC results were concordant as to the positivity, but only four SCLC had histological tissue available. Furthermore, all cases disclosed a larger percentage of positive tumor cells in surgical specimens, with one case also depicting a higher intensity. Although the differences were not very striking, they might be sufficient to impair the interpretation of the results, depending on the cutoff used, suggesting that INSM1 may be more suited for histology than cytology, as also disclosed by other studies [30–32,36]. Nonetheless, this study demonstrates that INSM1 is very useful for SCLC diagnosis in cytology samples, despite that cytology–histology correlation conclusions are hindered by a limited number of cases.

In the study by Rodriguez et al., a 97% (31/32) sensitivity for INSM1 ICC in cell blocks from 31 FNA and 1 bronchoalveolar lavage was reported, performing better than all other markers—63% (10/16) CGA, 78% (14/17) SYN, and 92% (22/23) CD56—and with perfect specificity for the differential diagnosis with NSCLC and other neuroendocrine tumor mimics, regarding morphology or clinical suspicion, such as acinar cell carcinoma, urothelial carcinoma, glomus tumor, clear cell sarcoma, and carcinoma NOS [34]. This study is limited by differences in the sample sizes of the various markers, making the comparisons less meaningful. Indeed, 84% (27/32) of the SCLC cases were strongly and diffusely positive for INSM1, and the only INSM1-negative case was also negative for all other markers, possibly on account of the extensive necrosis [34]. All four available surgical specimens were positive for INSM1, in agreement with the cytology specimens [34]. This study demonstrates that INSM1 is a reliable cytology marker, with a high sensitivity and specificity, having the advantage over CD56 of easier interpretation, as nuclear staining visualization is less challenging than membranous staining [34]. Subsequently, the same authors released another study, now focused on INSM1 in cytology samples of NE tumors in general, reporting 100% (32/32) sensitivity for SCLC (results not included in Table 1, as the cases came from a database of previously reported NE tumors, and likely, some may derive from the 2018 study) and also being expressed in pancreatic acinar cells and benign adrenal cells [22]. INSM1 was expressed in 90/91 neuroendocrine tumors, and interestingly, the case with negative expression, previously a poorly differentiated carcinoma with neuroendocrine features, was reclassified as SCLC after analysis of the surgical specimen, in which INSM1 tested positive, suggesting that INSM1 ICC may fail to match the IHC, as previously reported [22,30,31]. For a differential diagnosis with NSCLC, *INSM1* was 100% specific, and in other NE tumors, the expression was focal and weak, as depicted in a case of metastatic adrenocortical carcinoma [22]. Not focused on SCLC alone, this study is important, because it shows that INSM1 is also expressed in a wide variety of other neuroendocrine tumors.

Nakra et al. reported 91% sensitivity and 100% specificity for INSM1 ICC on direct smears in the diagnosis of SCLC vs. NSCLC, with an IHC/ICC concordance of 100%, although in a limited sample set (nine cases with both IHC and ICC available). In ICC-negative cases, the IHC results were not always concordant, and thus, a negative ICC for *INSM1* cannot accurately exclude SCLC, with other diagnostic markers required when morphology alone is not sufficient for a diagnosis [32]. INSM1 sensitivity in IHC was 97% (36/37), slightly superior to SYN—96% (27/28), and lower than CGA—100% (18/18) and CD56—100% (6/6), with the negative INSM1 case disclosing typical SCLC morphology and focal positivity for the other markers. These results might be considered unusual, since CGA commonly depicts a lower sensitivity than INSM1 [34–36]. Combining IHC and ICC, the INSM1 sensitivity was 97% (57/59). Nonetheless, the specificity was high, in agreement with previous studies [23,24,28,35–37], despite that its assessment was not the main focus and was only evaluated against NSCLC. Indeed, it seems critical to assess INSM1 staining patterns in tumors bearing close morphological patterns to SCLC, such as undifferentiated carcinomas, lymphomas, and round cell sarcomas [32], and to our knowledge, only one study has focused on that issue, concerning INSM1 [19]. The high sensitivity and specificity, alongside straightforward interpretation and strong nuclear staining, make INSM1 ICC an appealing diagnostic confirmation method of SCLC on cytology smears [32].

In another study, Viswanathan et al. disclosed an 89% sensitivity for INSM1 ICC in CB preparations, but as only nine cases were evaluated, these results were not included in Table 1 [30]. It remains important to analyze this study, however, as only a few studies with cytology–histology correlation are available [30–32,36]. Importantly, no expression was found in NSCLC, whereas LCNEC and carcinoids depicted a high expression. Surgical resections and TMA were also performed, disclosing 100% (8/8) *INSM1* positivity in SCLC cases using IHC, keeping a high expression in other pulmonary NE tumors, although not absolute (90% for carcinoids and 80% for LCNEC). This study is important from the perspective of specificity, as only 2% of NSCLC demonstrated ≥1 + INSM1 positivity in >5% of tumor cells among 511 cases, comprising SCC, ADC, non-NE large cell carcinoma (LCC), and other pulmonary tumors in histology samples. However, despite that *INSM1* expression was not observed in the cytology samples, only 15 cases of NSCLC (ADC and SCC) were available. Studies on a larger scale are important but quite difficult, as FNA samples provided are limited in materials. Interestingly, four out of five SCLC cases disclosed concordant staining in cytology and histology, with one case negative in cytology. Regarding the other markers, CD56, both in cytology and histology, depicted a better sensitivity (100%) but worse specificity (93%); SYN performed well, both in sensitivity—78% cytology and 100% histology—and specificity—100% in cytology and 91% in histology; and CGA had an excellent specificity (100%) but a poor sensitivity on both types of samples—22% cytology and 37.5% histology. The intra-tumoral variability of staining is a typical problem of immunochemical markers, especially on FNA samples, and, thus, markers with diffuse staining are key to ensure that positivity might be found across the whole tumor, which was the case for *INSM1* in this study [30]. Thus, the authors recommend the combination of *INSM1* with CD56 when dealing with cytology samples, implementing other markers as needed and based on tumor morphology.

Abe et al. assessed the INSM1 expression in bronchial brush cytology and in CB from pleural effusions, which remains, to date, the only study on this kind of sample [33]. The intensity of expression for INSM1 in bronchial brush CB was greater than that of synaptophysin, as disclosed by the higher Tumor Proportion Score (TPS) (45.18 vs. 32.50%), and there was a statistically significant correlation between the expression of those markers. The sensitivity reached 95.8%, with 100% specificity for *INSM1*, using a cutoff of 8.68% determined through the ROC curve analysis [33]. Notwithstanding the perfect specificity using that cutoff, INSM1 was focally expressed in 4/15 adenocarcinomas, emphasizing the need for a standard cutoff in practice, as those cases might have been deemed positive for *INSM1* expression with different experimental conditions, and only a few other studies have used statistically determined cutoffs [19,24,28,33]. In pleural effusion CB, all (five out of five) SCLC were INSM1+ (results not included in Table 1), with a TPS (62.98%) superior to bronchial brushing cytology, suggesting that the INSM1 cytology in those samples may be useful, although the sample size was too small to make any solid assumptions. This study, although of a small scale, highlighted the usefulness of INSM1 ICC for SCLC diagnosis in less frequently used types of samples, making way for more robust studies, as any type of sample can be valuable for a diagnosis, as well as emphasizing the need for a common cutoff value [33].

In a study evaluating the sensitivity of INSM1 in various NE tumors in direct smear samples, Hou et al. obtained a sensitivity of 91% for INSM1 immunocytochemistry in SCLC, superior to CGA (73%) and SYN (82%), although the conclusions were limited due to the very small number of cases (11) [29]. ICC was not performed in NSCLC, and in other lung NE tumors, 90% sensitivity was disclosed in 10 lung carcinoids. Only another study evaluating INSM1 ICC in direct smears was available in the literature [32], with a matching sensitivity but a larger number of cases tested. When all markers were combined (INSM1, SYN, and CGA), the sensitivity reached 100%. No data on CD56 ICC was available in both studies [29,32], which might be important to confirm the usefulness of this marker in direct smear samples. For SCLC, INSM1 staining was more diffuse and more intense than the two other markers, contrary to other NE tumors in general [29]. The authors considered INSM1

staining to be easier to interpret, even with a weak intensity, given the clear nuclear staining and minimal background staining frequently exhibited by cytoplasmatic markers [29]. They concluded that INSM1 should be the first-line marker when only one immunostaining marker was possible to perform in direct smear samples, but this conclusion should be considered with caution, as larger studies are needed to confirm it, as well as studies reporting the specificity of INSM1 ICC in this type of sample [29].

In summary, INSM1 is a useful marker for SCLC diagnosis in a variety of cytology samples, disclosing both a high sensitivity and specificity for the differential diagnosis with NSCLC, with the drawback also of showing expression in other types of NE tumors and even, to a lesser degree, in some types of NE cells. In general, studies have reported superior sensitivity to CGA and SYN, similar to that of CD56. In comparison to histology, the sensitivity was mostly similar or slightly inferior, with a good, although imperfect, matching between the cytology and histology samples, with sometimes slightly inferior intensity and extension, suggesting the superiority of INSM1 in histology over cytology. Although no consensus was reached regarding combined or single-marker use, the possibility of a marker that can save material and reduce costs is especially important for immunocytochemistry, and in the context of SCLC, further studies with larger casuistics, comparisons with other markers and tumor types and direct comparisons between histology and cytology use, should be encouraged.

### 4.4. INSM1 Gene Expression and Diagnostic Use

Few studies have reported on *INSM1* gene expression as a means for SCLC diagnosis. These have shown that *INSM1* gene expression correlates with neuroendocrine differentiation, being expressed in a variety of tumors, including insulinoma, glucagonoma, pituitary tumor, pheochromocytoma, medullary thyroid carcinoma, and, most importantly, SCLC, whereas the expression was not disclosed in non-neuroendocrine tumors or normal cells [9].

Lan et al. detected *INSM1* mRNA through a Northern blot analysis in 97% (30/31) of SCLC cases, whereas it was present in only 13% (4/30) of NSCLC cases [39] The case negative for *INSM1* mRNA expression also had undetectable CGA and L-Dopa-Decarboxylase (DDC) mRNA, both markers of neuroendocrine differentiation. In SCLC, *INSM1* mRNA was more sensitive than CGA and DDC—97 vs. 72 and 73%, respectively. Furthermore, four out of nine NSCLC positive for one of those NE markers also showed an expression of *INSM1* mRNA, and three carcinoids showed expression for all markers. The NE markers and mRNA expression were highly correlated in all cancers tested. These results suggest that the *INSM1* transcript is highly expressed in SCLC and, when present in NSCLC, may represent true neuroendocrine differentiation [39].

Pedersen et al. reported a high expression of *INSM1* mRNA in a microarray analysis and semiquantitative real-time PCR (RT-PCR) in most SCLC cell lines tested (85%) and an absent expression in normal tissues, whereas *INSM1* mRNA was not detected in NSCLC using RT-PCR [62,78].

A study aiming to characterize SCLC molecular profiles using cDNA microarrays found that a set of genes related to neuroendocrine differentiation, including *INSM1*, disclosed an altered expression, and that expression clearly differed between SCLC and NSCLC, further endorsing the use of this marker for differential diagnoses [74].

A subsequent study by Rosenbaum et al. found that the overexpression of *INSM1* mRNA quantified by quantitative RT-PCR (qRT-PCR) in gastrointestinal NE neoplasms was associated with more frequent tumor metastasis and that a higher expression correlated with strong IHC staining [21]. These data, although not obtained in SCLC, reveal that both *INSM1* mRNA and IHC expression may have diagnostic and prognostic value, simultaneously. Additionally, since stronger IHC staining was associated with higher gene expression, it strongly suggests an interdependence between mRNA and protein expression, so that both represent true NE differentiation [9,39].

The study by Staaf et al. answered one of the questions that Rosenbaum left open. They investigated the correlation between *INSM1* gene expression and the respective protein

by analyzing a total of 232 cases from two separate studies using RNA sequencing, and matching the mRNA results with the available IHC scores, revealing that *INSM1* gene expression was associated with NE histology in both studies, not only with *INSM1* IHC but also with SYN, CD56, and, to a lesser extent, with CGA [28]. They also investigated a set of 66 LCNEC from different molecular subtypes and found that LCNEC with the SCLC/SCLC-like subtype disclosed significant *INSM1* overexpression compared to other subtypes [28], suggesting that *INSM1*, even if present in many NE tumors, may be more specific of tumors with a SCLC genetic profile.

In summary, *INSM1* gene expression is associated with NE differentiation, directly correlates with *INSM1* IHC (protein) expression, and may have prognostic implications, being closely associated with the SCLC molecular profile. The sensitivity of genetic methods remains to be determined, since only one study disclosed any results in this regard [39]. With the advent of *INSM1* IHC, the potential of using alternative genetic sequencing techniques for this biomarker has moved into the background.

*4.5. INSM1 Diagnostic Use—Conclusions*

Unequivocally, *INSM1* is an important diagnostic biomarker of NE differentiation for SCLC, both in histology and cytology samples, depicting a high sensitivity, specificity, and reproducibility among studies. Its main advantage over other currently used marker is the exclusive nuclear staining and minimal background staining. Furthermore, *INSM1* is expressed both in primary SCLC and its metastases. The fact that INSM1 staining represents true NE differentiation is a double-edged sword, because expression means that stained cells have some level of neuroendocrine differentiation, which may potentially have future prognostic or therapeutic implications in cases of mixed SCLC tumors, but it also signifies that expression is found in other NE tumors, not only in the lungs but also in other locations. Thus, INSM1 expression alone does not allow for a reliable diagnosis of SCLC. Furthermore, INSM1 may also be expressed, albeit rarely, in NSCLC. The fact that no consensus in the positivity cutoff exists, either for histology or cytology, is also a limitation to its use and precludes a more accurate estimation of the sensitivity and specificity. Despite the high sensitivity, not all studies conclude the superiority of *INSM1* over the commonly used classic NE combined panel (SYN, CGA, and CD56), and thus, some authors do not endorse its use as a replacement. Nevertheless, the possibility of saving materials and reducing costs may justify INSM1 use as a standalone marker if differences in the sensitivity and specificity do not significantly differ following the definition of a universal cutoff value.

Comparing techniques, IHC for INSM1 seems to be slightly superior to ICC, both in sensitivity and specificity, but the differences are not marked enough to justify abandoning a commonly used method of SCLC diagnosis, considering the good histology–cytology correlation. As for INSM1 gene expression, there is concordance with IHC but no studies available comparing with *INSM1* ICC. Nonetheless, one may extrapolate that if cytology and histology marker expression are correlated, and the primary antibodies used are the same, it seems likely that the gene expression will also correlate with the ICC results. Importantly, INSM1 mRNA expression was shown, similar to IHC expression, to correspond to bona fide NE differentiation. Unfortunately, very few studies report on the use of *INSM1* gene profiling techniques, limiting comparisons.

We conclude that both INSM1 IHC and ICC are reliable ancillary methods for SCLC diagnosis and that standalone usage of the marker should be postponed until further studies are made and a standard cutoff for staining positivity is agreed on. Table 1 shows detailed results from a variety of studies on INSM1 immunochemical use for the diagnosis of SCLC, summarizing the above comments and showing results on the various sensitivities in INSM1 and classic NE markers, specificity for differential diagnosis with NSCLC, and INSM1 expression in other types of NE tumors of the lungs.

**Table 1.** *INSM1* immunochemistry for small cell lung cancer diagnosis—retrospective analysis since 2015.

| Study | No. of SCLC Samples | Type of Sample | Preparation | INSM1 Positive Cutoff Value | INSM1 Sensitivity, % (Positive Cases/ Total Cases) | CGA/SYN/CD56 Sensitivity, % (Positive Cases/ Total Cases) | Combined Classic NE Markers Sensitivity, % (Positive Cases/ Total Cases) | INSM1 Specificity (SCLC vs. NSCLC *), % | INSM1 Expression on Other Lung NE Tumors: LCNEC/ Carcinoids (Typical and Atypical), % (Positive Cases/Total Cases) |
|---|---|---|---|---|---|---|---|---|---|
| Fujino 2015 [4] | 27 | Resections | FFPE section | Any nuclear staining in tumor cells | 100 (27/27) | 70 (19/27)/ 63 (17/27)/ n/a | n/a | 100 | n/a |
| Fujino 2017 [38] | 19 | Resections | FFPE section | Any nuclear staining in tumor cells | 100 (19/19) | 74 (14/19)/ 58 (11/19)/ 68(13/19) | n/a | 100 | 100 (4/4)/ 100 (5/5) |
| Rooper 2017 [37] | 39 | Resections (TMA) | FFPE section | Any nuclear staining in tumor cells | 95 (37/39) | 49 (19/39)/ 62 (24/39)/ 70 (21/30) | 74 (29/39) | 96 | 91 (21/23)/ 100 (48/48) ** |
| Kriegsmann 2018 [35] | 144 | Resections (TMA) | FFPE section | ≥1% stained tumor cells in at least 1 of 2 cores per patient | 86 (124/144) | 74 (107/144)/ 85 (122/144)/ 92 (132/144) | 95 (137/144) | 99 | 42 (32/77)/ 79 (120/151) |
| Doxtader 2018 [36] | 41 | Transbronchial FNA, CT-guided aspirates and a fluid cytology specimen | Cellient cell blocks | Any nuclear staining in tumor cells | 93 (38/41) | 35 (14/40)/ 93 (37/40)/ 100 (40/40) | 100 (40/40) | 100 | 100 (1/1)/ 90 (9/10) |
| Rodriguez 2018 [34] | 32 | Aspirates (31) and bronchoalveolar lavage (1) | Cell blocks | Nuclear positivity in ≥1% tumor cells | 97 (31/32) | 63 (10/16)/ 78 (14/17)/ 96 (22/23) | n/a | 100 | n/a |
| Abe 2019 *** [33] | 24 | Bronchial brushing | Cell blocks | 8.68% cutoff determined through ROC curve analysis | 98 (23/24) | Only SYN: 86 (21/24) | n/a | 100 | n/a |
| Mukhopadhyay 2019 [23] | 64 | Small biopsies (32) and whole-tissue sections of resected tumor (32) | FFPE section | Any nuclear staining in tumor cells | 98 (63/64) | 83 (53/64)/ 100(64/64)/ 95 (61/64) | 100 (64/64) | 97 | 75 (18/24)/ 98 (63/64) |
| Švajdler 2019 [31] | 112 | Endoscopic biopsies (112) | n/a | Any nuclear staining in tumor cells | 81 (81/100) | Only CD56: 84 (84/100) | n/a | n/a | n/a |
| | | n/a—cytology sample (13) | Cell blocks (cytoblock) | | 54 (7/13) | Only CD56: 100 (13/13) | | | |

**Table 1.** *Cont.*

| Study | No. of SCLC Samples | Type of Sample | Preparation | INSM1 Positive Cutoff Value | INSM1 Sensitivity, % (Positive Cases/ Total Cases) | CGA/SYN/CD56 Sensitivity, % (Positive Cases/ Total Cases) | Combined Classic NE Markers Sensitivity, % (Positive Cases/ Total Cases) | INSM1 Specificity (SCLC vs. NSCLC *), % | INSM1 Expression on Other Lung NE Tumors: LCNEC/ Carcinoids (Typical and Atypical), % (Positive Cases/Total Cases) |
|---|---|---|---|---|---|---|---|---|---|
| Nakra 2019 [32] | 60 | Aspirates (36) | Direct smears | Any nuclear staining in tumor cells | 91 (30/33) | n/a | n/a | 100 | n/a |
| | | Small biopsies (37) | FFPE section | | 97 (36/37) | 100 (18/18)/ 96 (27/28)/ 100 (6/6) | | | |
| Hou 2020 [29] | 11 | Aspirates | Direct smears | Nuclear positivity in ≥ 5% of tumor cells | 91 (10/11) | 73 (8/11)/ 82 (9/11)/ n/a | n/a | n/a | Only carcinoids: 90 (9/10) |
| Narayanan 2020 [6] | 19 | Resections (TMA) | FFPE section | Any nuclear staining in tumor cells | 100 (19/19) | 80 (15/19)/ 100 (19/19)/ n/a | 100 (19/19) | n/a | 100 (9/9)/ 95 (83/87) |
| Sakakibara 2020 [24] | 78 | Resections (whole slides and TMA) | FFPE section | Calculated *H*-score of 5 | 92 (72/78) | 48 (37/78)/ 55 (43/78)/ 81 (63/78) | 85 (66/78) | 95 | 68 (30/44)/ 95 (18/19) |
| Staaf 2020 [28] | 24 | Resections (whole slides and TMA) and biopsies | FFPE section | Any nuclear staining in tumor cells | 92 (22/24) | 67 (16/24)/ 83 (20/24)/ 96(23/24) | n/a | 98 | 87 (20/23)/ 100 (7/7) |
| | | | | At least 10% positive tumor cells | 75 (18/24) | 46 (11/24)/ 79 (19/24)/ 88 (21/24) | 88 (21/24) | 99 | 61 (14/23)/ 100 (7/7) |

*INSM1* specificity (SCLC vs. NSCLC),% calculation: 100× (1—NSCLC *INSM1*-positive cases/total NSCLC cases); * Only adenocarcinoma and squamous cell carcinoma; ** 3 cases of LCNEC, and 2 carcinoids were thymic; *** pleural effusion samples were also evaluated, the results not included due to the small sample size (5 cases).

### 5. INSM1 as a Prognostic Marker

SCLC is characterized by its highly aggressive clinical behavior, frequent metastasis, and high recurrence rate after the initial response to first-line chemotherapy [2–4]. Given its dismal prognosis, there is a high demand for biomarkers that may predict patient outcomes, but no consensus has been reached concerning the prognostic biomarkers for lung NEC, including SCLC [6]. It was shown that classic NE markers such as CGA, SYN, and CD56, typically used as ancillary immunohistochemical markers for NE tumors diagnosis, may also have prognostic value [4,79,87,88]. Naturally, with the discovery of new NE markers, alternative uses to diagnoses were investigated. In 2015, Fujino et al. [4,38] reported that *INSM1* knockdown in SCLC cell lines resulted in the suppression of cell proliferation and increased apoptosis, suggesting that *INSM1* may have an oncogenic role in SCLC progression. Following this observation, subsequent studies have attempted to demonstrate the prognostic value of *INSM1* [24,89–92].

McColl et al. aimed to better understand the mechanisms underlying chemorefractory SCLC. Using gene profiling techniques, two subgroups of SCLC were identified: one predominantly expressing *INSM1* mRNA and the other expressing mostly Yes1-associated transcriptional regulator (YAP1), a key mediator of the Hippo pathway [92]. Most SCLC cases analyzed disclosed high *INSM1* expression and low YAP1 expression. Western blotting showed that, at the protein level, differences were maintained, and there were even 15 SCLC cell lines disclosing mutually exclusive expression of the markers. The IHC staining of both markers also corroborated these findings, with the majority of tumors showing strong *INSM1* staining, whereas YAP1 was positive in only a few. Then, *INSM1* IHC staining in 55 additional specimens of SCLC (in a TMA) disclosed that a higher *INSM1* expression was associated with an increased overall survival (OS), progression-free survival (PFS), and chemo-responsiveness [92], indicating that a higher *INSM1* expression may be associated with a better outcome, contrarily to Fujino et al.'s hypothesis [4,38]. McColl et al. proceeded to investigate subgroups' responses to commonly used chemotherapy agents for the treatment of SCLC, and all three agents (cisplatin, etoposide, and irinotecan) were more effective against cells from the *INSM1*-expressing group. They also demonstrated that BCL2 apoptosis regulator (*BCL2*), a gene associated with a better prognosis expressed in the majority of SCLC [93], was expressed at both the transcript and protein levels in cells from the aforementioned group [92], suggesting a connection between both genes. The authors proposed that SCLC might be prognostically divided into two groups: the most prevalent "classic" SCLC, which is *INSM1*+ and shows a relatively better prognosis, and a variant group, *INSM1*-, in which YAP1+ tumors are included, disclosing a worse prognosis [92]. Despite that the *BCL2* expression was increased in the first group, the precise mechanisms associated with chemosensitivity and a better outcome remain elusive. This study shows that *INSM1* may be useful for stratifying patients into groups susceptible or resistant to chemotherapy, with potential therapeutic implications in the future.

Conversely, Minami et al., in a study comprising 75 patients with SCLC and LCNEC, aimed to determine the relationship between *INSM1* expression (through IHC) and prognosis. They found that the OS and recurrence-free survival (RFS) were significantly worse not only in the *INSM1*+ group but also in *INSM1*+ SCLC patients [90]. Moreover, patients with clinical stage I and *INSM1*+ SCLC endured significantly poorer RFS than the *INSM1*- patients with the same disease stage. Thus, these authors proposed that the first group of patients should be managed more aggressively, with preoperative chemotherapy, whereas the second would benefit from surgery alone. Other NE markers depicted no differences in the OS or RFS, except for SYN positivity, which was associated with a significantly poorer RFS. After a multivariable analysis, taking into account multiple factors associated with the OS and RFS, *INSM1* was shown to be the strongest predictor of a poor prognosis for the OS and RFS [90]. These results support the use of *INSM1* as a prognostic biomarker and emphasize its superiority over other NE markers. The findings of this study are, however, limited by the relatively small number of subjects and by the retrospective nature of the study. Furthermore, a prognosis might have been underestimated owing to the inclusion of

patients that did not go through postoperative chemotherapy due to poor respiratory or overall function [90].

Sakakibara et al. conducted a study aiming to determine the prognostic role of INSM1 in SCLC [24]. The SCLC subgroup with absent IHC NE marker expression disclosed a statistically significant better prognosis, in accordance with their previous study, compared to the group with NE marker expression, and within that group, inferior expression levels correlated with a better outcome. As for INSM1 IHC expression, a subgroup with low *INSM1* expression had a statistically better prognosis than the subgroup with high expression; therefore, low INSM1 and low NE markers expression are possibly associated with a good prognosis [24]. Although, in the univariable analysis, INSM1 and NE expression was statistically associated with lower survival, the multivariable analysis disclosed only marginal significance. This study further alerts to the possibility of using INSM1 to guide the therapy and follow-up of SCLC patients, although limited through the inclusion of some cases that had pre-op chemotherapy [24,90].

In a different way, Baine et al. aimed to identify different SCLC subtypes based on ASCL1, NEUROD1, POU2F3, and YAP1 IHC expression, providing a histopathologic and immunohistochemical characterization, in which the expression of NE markers was included [91]. Regardless of this study not directly focusing on INSM1, it is important because these subtypes may be correlated with previously INSM1-defined SCLC subtypes [92]. The NE expression, including INSM1, was markedly reduced in the ASCL1/NEUROD1 double-negative group and was inversely associated with the YAP1 expression. The YAP1 expression seemed to be associated with combined SCLC histology, and a higher expression was observed in the NSCLC component. Most SCLC were found to express *ASCL1* and/or NEUROD1, which were associated with a high NE program [91]. In line with McColl et al., this study shows that SCLC may be divided into two distinct groups: one with high ASCL1 and/or NEUROD1 levels, with concomitant high INSM1 expression, and another with a low expression of those two markers, which, in turn, shows low INSM1 expression [91,92]. Contrarily to McColl et al., this study considered YAP1 expression in SCLC to be mostly irrelevant, and thus, no subtype was defined with that marker [91,92]. One important finding of this study is that even the ASCL1/NEUROD1 double-negative group showed NE expression to a lesser extent, raising doubts about the "non-NE subtype of SCLC" [94], as the availability of more sensitive markers such as INSM1 make this classification obsolete [23,37,91]. Despite no analysis of the patient prognostic factors being performed, this study was useful to corroborate previous findings that further confirmed the INSM1 prognostic value of INSM1 [91,92]. Some studies have already depicted a possible relation between INSM1 and ASCL1 in SCLC [4,12], but more are needed concerning the other markers, as new molecular pathways may help find new treatment strategies and improve the overall knowledge of this particular type of cancer.

In a recent study from Xu et al., researchers examined *INSM1* IHC expression in 73 SCLC samples from surgical resections and reported that a high INSM1 expression was positively correlated with lymph node metastasis (LNM) and advanced TNM stage [89]. In a univariable analysis of the patient prognosis, the high INSM1 group showed worse overall survival, and the multivariable analysis deemed INSM1 as an independent prognosticator for SCLC. In the group of patients undergoing chemotherapy with LNM, INSM1 positivity was significantly associated with reduced OS and chemoresistance, but this was not disclosed in the group of patients undergoing chemotherapy without LNM. Interestingly, the INSM1 status did not seem to affect the prognosis in patients who did not undergo chemotherapy. The researchers went further and investigated INSM1's non-neuroendocrine role in SCLC using in vitro experiments to illuminate the possible mechanisms underlying those findings. They found that INSM1 overexpression downregulated protein kinase AMP-activated catalytic subunit alpha 2 (PRKAA2) expression, reduced glucose intake activity, augmented tumor migration capacity, and induced cisplatin resistance [89]. These effects, both in vitro and in vivo, were reversed by metformin, an antidiabetic drug that currently is under study as an adjuvant for chemotherapy in lung cancer patients [95–99].

Thus, INSM1 might be useful beyond diagnostic and prognostic use to help select cancer patients for trials with metformin [89]. An advantage of this study is that all selected patients had not received any prior treatment, a factor that may have contributed to bias in other studies [24,89]. Nevertheless, the study had some limitations: (1) a modest sample size; (2) a failure to confirm the increased tumor growth with INSM1 overexpression in vitro, which should be expected based on the in vivo results; and (3) the use of a single SCLC cell line. In short, this study showed that INSM1 may predict the poor prognosis in SCLC, given its contribution to tumor metastasis and chemoresistance, and these effects might be explained by PRKAA2 downregulation [89]

In summary, INSM1, an undisputedly useful diagnostic marker, is still in its infancy concerning a prognostic role in SCLC. At this point, it is quite difficult to claim whether it will be used to predict aggressive or indolent behavior, since, despite most studies affirming that *INSM1* high expression associates with the overall worse prognosis, there was one study with opposed results [92], making meaningful conclusions impossible with so few studies on this matter. Efforts to define SCLC subtypes should be continued, as there is no current consensus about their definition and might simplify the search for prognostic factors. Focus on molecular pathways involving *INSM1*, and known oncogenes or tumor suppressor genes should also be useful and could serve as an entry point for the discovery of novel therapeutic targets.

## 6. Gene Therapy and Other Treatment Options Targeting INSM1 or Related Molecules

Compared to NSCLC, SCLC always had a more limited range of therapeutic options [75]. Cisplatin and etoposide are used as a first-line chemotherapy for many years, with initial chemosensitivity but later developing resistance, and only recently, immunotherapy in the form of nivolumab, a PD-1 antagonist, was approved for third-line use by the FDA in recurrent SCLC, unfortunately with less than 20% of patients benefiting from this strategy [89,94,100,101].

*INSM1*'s very low expression in adult tissues and re-expression in neuroendocrine tumors, disclosing a high specificity, has made it an attractive target for novel therapies. No studies directly targeting *INSM1* have been carried out, as the transcription factors are especially difficult to target using small molecules [102,103]. Instead, the focus has been placed on targeting *INSM1*-expressing tumors using *INSM1* promoter-specific suicide gene therapy [62–68] or the molecules involved in *INSM1*-related signaling pathways [77,102–104].

In transcriptionally targeted suicide gene therapy, a suicide gene contained in a vector, usually of a viral nature, is incorporated into cancer cells, and through a cancer-specific promoter, its activity is regulated, allowing the conversion of a nontoxic prodrug—for example, ganciclovir—to a cell toxic agent, with the goal of avoiding the side effects produced by chemotherapy agents in noncancer cells [64]. It is long-known that *INSM1* contains a promoter region with the genes responsible for regulating its own expression [105]. Published in 2006, a study by Pedersen et al. was the first to report *INSM1* usefulness in targeted suicide gene therapy in SCLC [62]. The authors found that a 1.7-kb region in the *INSM1* promoter showed a high expression in SCLC cell lines but was absent in non-neuroendocrine cell lines, with sufficient activity and specificity for inclusion in gene therapy and superiority vs. other contemporarily available promoters. When regulated by this promoter region, the suicide gene herpes simplex virus thymidine kinase (*HSV-TK*) in combination with ganciclovir significantly augmented the sensitivity for the prodrug in SCLC cell lines. Considering these discoveries, the *INSM1* promoter has been suggested as an exciting new promoter for gene therapy given the very high activity and tumor specificity, propelling further studies [62]. These findings were replicated both in vitro and in vivo in primitive neuroectodermal tumors that show neuroendocrine features similar to SCLC [63]. In another study, *INSM1* promotor-driven adenoviral *HSV-TK* gene therapy was tested in the treatment of various neuroendocrine tumors, and it was found that the adenoviral genome interfered with the promoter specificity, resulting in in vivo off-target activity in mouse pancreas, kidneys, and lungs [66]. An interesting feature of suicide gene



therapy is the possibility of multiple modifications, and in their studies, Akerstrom et al. demonstrated that, by adding a chicken β-globin HS4 insulator sequence and two copies of mouse neuronal-restrictive silencer element (NRSE), not only nonspecific *INSM1* promoter activation was reduced, but also, the activity of the new construct was increased compared to the original, both in vitro and in vivo, although only in vitro for SCLC [66,67]. Subsequently, the modified *INSM1* promoter-driven adenoviral *HSV-TK* construct in combination with ganciclovir was tested in insulinoma in vitro and in vivo, with results analogous to those of previous studies [68]. Studies in other types of tumors also proved useful, since the underlying basis of the therapeutic method extends to the various *INSM1*-expressing neuroendocrine tumors. Moreover, the same research team conducted a study aimed at detecting neuroendocrine tumors, including SCLC, using a modified *INSM1* promoter in an adenoviral construct combined with the *Gaussia* luciferase gene. They found that, after the infection of an *INSM1*-positive NE lung tumor in a xenograft mouse model, luciferase activity was detectable, demonstrating a less conventional method of diagnosing these tumors and an alternative use for adenoviral constructs [106]. Christensen et al. demonstrated the efficacy of *INSM1* promoter-based suicide gene therapy both in vitro and in vivo in SCLC cell lines and xenografted mice, using a YCD/5-FC and a YCD-YUPRT/5-FC model. The therapy proved to be highly cytotoxic to SCLC but not to other cell lines and significantly delayed the tumor growth in SCLC xenografts compared to the controls, with better results than the previous HSV-TK/ganciclovir model, mainly due to YCD-YUPRT-produced toxins having an important bystander cytotoxic effect [64]. Later, the *INSM1* promoter-driven YCD-YUPRT/5-FC suicide gene therapy model was proven efficient in SCLC chemoresistant cell lines and xenografts, even in cases displaying higher *INSM1* promoter activity compared to chemosensitive SCLC, notwithstanding the high levels of cytotoxicity in both. In the same study, another *INSM1* promoter-driven construct was tested—NTR with the SN27686 prodrug—also with positive results. Interestingly, *INSM1* promoter-driven YCD-YUPRT/5-FC gene therapy combined with chemotherapy, as well as double suicide gene therapy, disclosed superior results to single gene therapy in chemoresistant SCLC cells [65].

In more recent years, the focus of *INSM1*-related therapeutic strategies has shifted away from gene therapy and has moved towards targeting molecules being involved in signaling pathways in which it plays a role. In the anterior pituitary gland, *INSM1* interacts with LSD1 through its SNAG domain at the N-terminus, contributing to neuroendocrine cell differentiation [69]. LSD1 inhibitor T-3775440 has been shown to efficiently suppress SCLC proliferation in vitro and delay tumor growth in vivo, possibly by interfering with the *INSM1*-LSD1 interaction and preventing the expression of NE genes, such as *ASCL1*. Through *INSM1* knockdown, comparable results in gene expression and cell proliferation were obtained, sustaining those findings [102]. This study supports the hypothesis that SCLC carcinogenesis is, in part, led by neuroendocrine differentiation, as LSD1 interactions with *INSM1* and the consequent effects are similar to those observed in the anterior pituitary gland, opening the way to new therapeutic targets. As previously discussed in the introduction section, *INSM1* is involved in the *Shh* and *MYCN* common pathways [77]. *Shh* can upregulate *MYCN* via activation of the *PI3K/AKT* pathway or via Smoothened activation. Different Shh or related molecules inhibitors were used to treat SCLC cells, and it was found that 5E1, a Shh direct inhibitor, and GANT-61, an inhibitor of Gli1, a transcription factor induced by Smoothened that directly upregulates *MYCN*, downregulated the *INSM1* expression and suppressed SCLC proliferation in single uses, with GANT-61 exhibiting a superior effect but requiring an above threshold dosage. Although Smoothened inhibitors such as cyclopamine or vismodegib did not affect cancer growth when used alone, they amplified the inhibitory effects when combined with 5E1 or GANT-61, increasing the sensitivity of the latter and enabling its use within a safe therapeutic range (10 μM). Indeed, GANT-61 plus vismodegib was the most efficient combination. These findings support the role of *INSM1* as a crucial SCLC oncogenic factor, since *INSM1* knockdown inhibited the tumor growth and enhanced the effects of Shh pathway inhibitors. Apoptosis via caspase-3 activation was depicted in both *INSM1* knockdown and Shh pathway inhibitors [77]. *Shh*

signaling pathway inhibitors represent an exciting new opportunity for SCLC treatment, albeit further studies with different inhibitor combinations are required to corroborate these findings. The same authors reported that 5′-Iodotubercidin repressed *INSM1* expression, restricting the growth of human neuroblastoma cell lines. The main mechanism involved was *ADK* blockage, but the suppression of the ERK$^{1/2}$ and AKT signaling pathways was also present, suggesting a hypothetical use in SCLC, since these are shared pathways [104]. A recent study by Norton et al. found through genome-scale CRISPR/Cas9 inactivation screening to murine SCLC cell lines that protein neddylation inhibition was effective in suppressing those cells [103]. MLN4924 effectively caused cell death in vitro and in vivo and suppressed the expression of master neuroendocrine regulators such as *ASCL1* and *INSM1*. The authors could not identify the molecular mechanism underlying these results but hypothesized that treatment stabilized a putative *INSM1* transcriptional repressor regulated by ubiquitin-mediated proteolysis, either directly or through neddylation of an upstream regulator of that repressor [103]. Thus, SCLC treatment strategies directed to neuroendocrine differentiation controllers maintain their relevance.

In conclusion, suicide gene therapy and targeting related molecules are both effective strategies of exploiting *INSM1* for SCLC treatment. Over the years, modifications have been made to *INSM1* promoter-driven gene therapy that increased its sensitivity toward cancer cells and decreased the collateral damage to normal cells. *INSM1* promoter was used to regulate gene therapy in three models: (adenoviral) HSV-TK/ganciclovir, YCD or YCD-YUPRT/5-FC, and NTR/SN27686. Further studies with direct comparisons are required. More recently, research has focused on targeting *INSM1*-related molecules, namely LSD1, Shh, and *MYCN* signaling pathway molecules, and protein neddylation. *INSM1* and many of these molecules are related to the induction of NE differentiation in SCLC, which is increasingly recognized as a key factor in oncogenicity. Finding new ways to control neuroendocrine cell states will be crucial for developing new tools against this and other types of neuroendocrine neoplasms. Lastly, targeting *INSM1* directly is a difficult challenge but may prove largely beneficial. A summary of the different treatment options in SCLC involving *INSM1* is displayed in Figure 2.

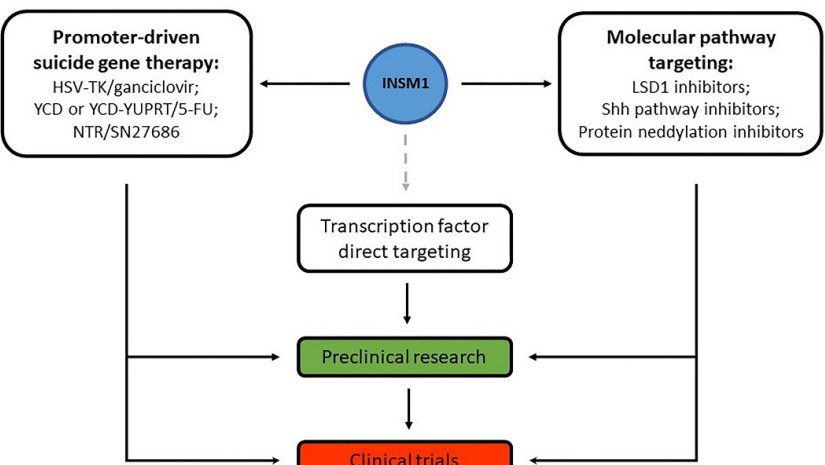

**Figure 2.** INSM1 treatment options in SCLC. Summary of the researched therapeutic approaches for SCLC using *INSM1*. Currently, there are no studies available on *INSM1* as a direct target, due to the difficult nature of targeting transcription factors [102,103]; therefore, preclinical trials would be of high interest. *INSM1* has been used as a regulator of suicide gene therapy in HSV-TK/ganciclovir, YCD or YCD-YUPRT/5-FU, and NTR/SN27686 models [62–68]. Alternatively, targeting multiple molecules in *INSM1*-related pathways has also been an effective approach for SCLC treatment. LSD1 inhibitors, Shh pathway inhibitors, and protein neddylation inhibitors successfully reduced the SCLC tumor growth, and 5′-Iodotubercidin was efficient in neuroblastoma treatment (not shown in the figure) [77,102–104]. These studies were conducted in in vitro and in vivo in mouse models; thus, clinical trials may be the next step to establishing their effectiveness.

## 7. Conclusions

SCLC represents one of the most aggressive and difficult forms of cancer to treat. Furthermore, its diagnosis is made more complex owing to the difficulty in obtaining representative and high-quality samples, given its usual central perihilar location. It is included in the extensive group of neuroendocrine neoplasms, which share the hallmark expression of neuroendocrine markers. Although not essential for SCLC diagnosis (as typical morphological features may suffice), diagnostic biomarkers can prove very beneficial to aid in diagnosis, especially in dubious cases, such as in the not-uncommon presence of tissue necrosis. In this review, we characterized a molecule—*INSM1*—that may positively respond to the challenges in SCLC, disclosing diagnostic, as well as possible prognostic and therapeutic, applications. *INSM1* has emerged as a diagnostic NE biomarker for SCLC since 2015, primarily as an IHC or ICC marker. Although CGA, SYN, and CD56 are widely used immunochemical markers but all of them have limitations and often have to be used simultaneously in a triple marker panel, leading to the extensive use of limited tissue. *INSM1* is a reliable IHC and ICC marker with high sensitivity, specificity, and a distinctive exclusively nuclear expression. *INSM1* sensitivity was disclosed in the majority of the studies as superior to CGA and SYN, as well as similar to CD56 and combined triple marker use, with a universally high specificity for the differential diagnosis with NSCLC. Staining represents true NE differentiation, which is an advantage in the differential diagnosis with NSCLC, despite some studies reporting a rare expression in that type of tumor [27], but a disadvantage in comparisons with other NE neoplasms. Importantly, the expression is retained in metastasis, facilitating a diagnosis in some cases. Nonetheless, no consensus has been reached concerning *INSM1* standalone use and a standard cutoff for immunohistochemistry. Larger-scale studies to address these issues are a priority. Controversy persists regarding *INSM1* and SCLC prognosis, and further studies are necessary to clarify this matter. Suicide gene therapy using the *INSM1* promoter as a regulator in different models and targeting molecules in *INSM1*-related signaling pathways, such as LSD1, Shh, and protein neddylation, disclosed successful results in vitro and in vivo in SCLC cell lines and xenografted mouse models. The next steps are to find new ways to alter the neuroendocrine differentiation of SCLC and to conduct clinical trials with these therapeutics. Direct targeting of *INSM1* or transcription factors in general is a difficult challenge but may result in substantial survival improvement in SCLC patients. We may thus conclude that the transcription factor *INSM1* represents a valuable biomarker for SCLC diagnosis that also provides ample opportunities for the development of new prognostic and therapeutic strategies.

**Author Contributions:** Conceptualization, R.R. and R.H.; writing—original draft preparation, R.R.; and writing—review and editing, R.H. All authors have read and agreed to the published version of the manuscript.

**Funding:** This research received no external funding.

**Institutional Review Board Statement:** Not applicable.

**Informed Consent Statement:** Not applicable.

**Data Availability Statement:** Not applicable.

**Acknowledgments:** The authors would like to thank José Pedro Sequeira for the assistance in the preparation of the figures and manuscript formatting.

**Conflicts of Interest:** The authors declare no conflict of interest.

## Abbreviations

*ASCL1*, achaete-scute family BHLH transcription factor 1; CGA, chromogranin A; Gli1, GLI family zinc finger 1; *Hes1*, hes family bHLH transcription factor 1; *INSM1*, insulinoma-associated protein 1; *MYCN*, *MYCN* proto-oncogene bHLH transcription factor; *NCAM1*, neural cell adhesion molecule 1 (CD56); *Notch1*, notch receptor 1; POU3F2, POU class 3 homeobox 2; SCLC, small cell lung carcinoma; *Shh*, sonic hedgehog; SYN, synaptophysin; CGA, chromogranin A; CT, computed tomography; FFPE, formalin-fixed paraffin-embedded; FNA, fine needle aspiration; *INSM1*, insulinoma-associated protein; LCNEC, large cell neuroendocrine carcinoma; n/a—not available; NE, neuroendocrine; NSCLC, non-small cell lung cancer; SCLC, small cell lung carcinoma; SYN, synaptophysin; TMA, tissue microarray.

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
