# Peer review of "Insulinoma-Associated Protein 1 (INSM1): Diagnostic, Prognostic, and Therapeutic Use in Small Cell Lung Cancer"

_jmp, doi:10.3390/jmp3030013_

Round 1

Reviewer 1 Report

The manuscript entitled "Insulinoma-associated protein 1 (INSM1): diagnostic, prognostic, and therapeutic use in small cell lung cancer" highlighted that  INSM1 represents a valuable biomarker for SCLC diagnosis that additionally offers vast opportunities for the development of new prognostic and therapeutic strategies.

The review is really weel written and of interest for the audience.

Minor comments:

- The Authors should provide the expand forms for all acronyms, including gene acronyms, through the text when they first appear.

- Gene acronyms should be written in italics.

- The Authors should provide high quality figures.

Author Response

The manuscript entitled "Insulinoma-associated protein 1 (INSM1): diagnostic, prognostic, and therapeutic use in small cell lung cancer" highlighted that INSM1 represents a valuable biomarker for SCLC diagnosis that additionally offers vast opportunities for the development of new prognostic and therapeutic strategies.

The review is really well written and of interest for the audience.

Re: We thank the positive comments of the Reviewer

Minor comments:

- The Authors should provide the expand forms for all acronyms, including gene acronyms, through the text when they first appear.

Re: We thank the Reviewer for calling our attention to this. The text was corrected throughout.

- Gene acronyms should be written in italics.

Re: We thank the Reviewer for calling our attention to this. The text was corrected throughout.

- The Authors should provide high quality figures.

Re: We thank the Reviewer fpr calling our attention to this. The quality of the figures was enhanced.

Reviewer 2 Report

The authors provide a nice overview and summary of published studies for INSM1. The review is comprehensive in scope, and should prove to be a useful reference. A few points:

1-It seems odd that the authors do not provide any histologic/cytologic images demonstrating INSM1 expression in small cell carcinoma (as well as the limited neuroendocrine expression in normal lung), possibly comparing to Chromogranin, Synaptophysin, and CD56. A histologic/cytologic figure would improve the paper.

Minor points:

Line 14: would eliminate the phrase "difficult to diagnosis"

Line 46: since this not mentioned anywhere else in the paper, would be important to provide the WHO criteria on mitotic count for resection specimens to diagnose small cell carcinoma: >10/2mm2 mitoses. Although not a stated criteria, authors could also mention high Ki-67/MIB-1 proliferation marker expression is useful in small biopsy specimens.

Line 307: should add "%" for numbers 49, 62, and 70

Line 362: 63/64=98% (not 100%)

Line 569: should add the number of cases for CD56 (% listed at 92%)

Line 1050: should add space between dash and INSM1: ... - INSM1 - ...

Author Response

The authors provide a nice overview and summary of published studies for INSM1. The review is comprehensive in scope, and should prove to be a useful reference. A few points:

1-It seems odd that the authors do not provide any histologic/cytologic images demonstrating INSM1 expression in small cell carcinoma (as well as the limited neuroendocrine expression in normal lung), possibly comparing to Chromogranin, Synaptophysin, and CD56. A histologic/cytologic figure would improve the paper.

Re: We thank the Reviewer for the positive comments. The idea underlying this paper was to introduce INSM1 immunohistochemistry.into our routine practice, depending on the systematic analysis of published results. Thus, at this point, we have no experience with INSM1 immunostaining, so that we can not provide the pictures. Nonetheless, the references cited in the paper contain representative images which the reader may choose to view.

Minor points:

Line 14: would eliminate the phrase "difficult to diagnosis"

Re: We thank the Reviewer for the suggestion. The text was changed accordingly.

Line 46: since this not mentioned anywhere else in the paper, would be important to provide the WHO criteria on mitotic count for resection specimens to diagnose small cell carcinoma: >10/2mm2 mitoses. Although not a stated criteria, authors could also mention high Ki-67/MIB-1 proliferation marker expression is useful in small biopsy specimens.

Re: We thank the Reviewer for the suggestion. This was added to the text.

Line 307: should add "%" for numbers 49, 62, and 70

Re: We thank the Reviewer for the suggestion. The text was changed accordingly.

Line 362: 63/64=98% (not 100%)

Re: We thank the Reviewer for the suggestion. The text was changed accordingly.

Line 569: should add the number of cases for CD56 (% listed at 92%)

Re: We thank the Reviewer for the suggestion. The text was changed accordingly.

Line 1050: should add space between dash and INSM1: ... - INSM1 - ...

Re: We thank the Reviewer for the suggestion. The text was changed accordingly.